



# Brief communication: New sonde to unravel the mystery of polar subglacial lakes

Youhong Sun[1,2], Xiaopeng Fan[1], Bing Li[2], Yuansheng Li[3], Guopin Li[4], Haibin Yu[5], Hongzhi Li[6], Dongliang Wang[7], Nan Zhang[1], Da Gong[1], Rusheng Wang[1], Yazhou Li[2] and Pavel G. Talalay[1]

[1]Polar Research Center, Jilin University, Changchun, China
   [2]China University of Geosciences, Beijing, China
   [3]Polar Research Institute of China, Shanghai, China
   [4]Nanjing Institute of Astronomical Optics and Technology, National Astronomical Observatories, CAS, China
   [5]College of Electronics and Information, Hangzhou Dianzi University, China
[6]National Ocean Technology Center, Tianjin, China
   [7]Aerospace System Engineering Shanghai (805), Shanghai, China

*Correspondence to*: Pavel G. Talalay (ptalalay@yahoo.com) and Xiaopeng Fan (fxp@jlu.edu.cn)

**Abstract.** The newly developed RECoverable Autonomous Sonde (RECAS) allows sampling and analysis of subglacial water while the subglacial lake remains isolated from the surface. The sonde was successfully tested in East Antarctica during 2021-

2022 field season: it reached the ice sheet base at the depth of 200.3 m, sampled melted water from basal ice and recorded the parameters (pressure, temperature, pH, and conductivity) of melted water. Then the sonde returned to the ice surface. The average downward penetration rate was 1.85 m h$^{-1}$, and the average upward penetration rate was 2.94 m h$^{-1}$.

## 1 Introduction

The base of Antarctic and Greenland ice sheets, polar ice caps, and valley glaciers demonstrates alternating warm-based and

cold-based areas. According to a hybrid ice sheet/ice stream model approximately 20% of Antarctic ice sheet is likely frozen to the bed, other larger area of the ice sheet base is under the pressure-melting point (Pattyn, 2010). The melted water at the base would eventually collect in topographic hollows and trenches forming complicated subglacial hydrological system.

The presence of a vast network of lakes, rivers, and streams, is verified by remote-sensing surveys in which flat reflectors at the bottom of ice sheet were interpreted as indicating subglacial accumulations of liquid water. The latest global inventory

identified 773 subglacial lakes in total, which includes 675 from Antarctica, 64 from Greenland, 2 beneath the Devon Ice Cap, 6 beneath Iceland's ice caps and 26 from valley glaciers (Livingstone et al., 2022). This will increase as surveys improve spatial coverage. The hydraulic potential reveals that some of the lakes could be hydraulically connected with one another and with the ocean, whereas others may not be in communication. Sealed from the Earth's atmosphere for many years, subglacial aquatic environment, especially hydraulically disconnected systems, may provide unique information about microbial

evolution, the past climate of the Earth, and the formation of the ice sheets and glaciers (Pearce, 2009; Siegert et al., 2016). The discovery of subglacial aquatic environments has opened an entirely new area of science in a short period of time.



Although the modern observations widely employ remote-sensing instruments to provide indirect indications of subglacial environment phenomena, the direct observation and sampling by drilling are still much needed for hydrological, chemical and microbiological studies. Drilling down and retrieving clean samples from deep subglacial lakes would provide scientists with

a treasure trove comparable to the lunar rocks, giving insight into the origins and evolution of our planet. Currently, four attempts were made to access, to measure in situ properties and to directly sample subglacial Antarctic lake environments.

In February 2012, Russian team first penetrated into subglacial Lake Vostok, largest subglacial lake in Antarctica, at a depth of 3769.3 m, allowing lake water to enter and freeze within the lower part of the ice-core borehole, from which further coring recovered a frozen sample of surface lake water (Lukin and Vasiliev, 2014). Unfortunately, when the subglacial water entered

the borehole, it mixed with the toxic drilling fluid, resulting in contamination of the subglacial water samples (Alekhina et al., 2017).

Second, UK team tried to deploy a hot-water drill and to sample the water column and sediments of subglacial Lake Ellsworth approximately 3000 m beneath the surface of the West Antarctic ice sheet (Siegert et al., 2014). Attempts to access this subglacial lake were unsuccessful in December 2012. The main reason for the failure relates to a subsurface cavity of water

300 m beneath the ice surface that could not be connected to by the main drill hole, and which consequently resulted in insufficient water available to continue drilling deeper.

Third, in early 2013, and fourth, in the season of 2018-2019, US teams successfully drilled with hot water circulation into subglacial lakes Whillans and Mercer, hydraulically active lakes at the coastal margin of West Antarctica (Priscu et al., 2021; Tulaczyk et al., 2014). However, during penetration and sampling subglacial lakes were connected to the surface via a borehole.

In addition, the ice thickness above these lakes is quite shallow (800-1067 m) leaving the problem of clean sampling of the deep subglacial lakes still unsolved.

In 2019, a joint UK-Chile collaborative partnership was started for the exploration of subglacial Lake Centro de Estudios Científicos (CECs) at the depth of near 2650 m applying also deep hot water drilling system (Makinson et al., 2021). Current plans included comprehensive field testing during the 2021–2022 austral summer at a site near targeted subglacial lake.

Unfortunately, because of COVID19 issues UK-Chile subglacial lake CECs project has been cancelled (K. Makinson, pers. comm., 2021).

The polar engineering community continues to work on developing of the new concepts and technologies for sampling subglacial lakes (Blake and Price, 2002; Fleckenstein and Eustes, 2007). However, these methodologies are still under preliminary concept level. In this paper we present the newly developed technology to sample subglacial lakes while the

subglacial lake remains isolated from the surface and results of the first field tests.

## 2 Methods

To sample subglacial lakes, we proposed to use the freezing-in hot-point thermal drill (so-called Philberth probe) that is able to move towards the glacier base by ice melting while the melted water refreezes behind the probe. This type of the probe was



proposed by Phillberth (1976) to study the temperature distribution in ice sheets. The most outstanding characteristic of the
probe is that the wires used for the transmission of electrical power to them and the signals from them pay out of the advancing
drill and became fixed in the refreezing meltwater above it. The probe only travelled one way. In the summer of 1968, the
Philberth probe reached the remarkable depth of 1005 m at station Jarl-Joset in Greenland.

In the following years, nine similar freezing-in thermal probes were designed in different conceptual and testing phases but all
of them travelled one way (probes are reviewed in Talalay, 2020). Our proposed freezing-in probe – RECoverable Autonomous
Sonde (RECAS) – is equipped with two electrically powered melting tips located at the upper and lower ends of the sonde
(Talalay et al., 2014). Thus, it can drill downwards and upwards and move in the borehole using inner cable recoiling
mechanism on the same way as a spider climbs on the silk line (Fig. 1). The sonde intends to explore subglacial aquatic
environments with minimal chemical and microbial contamination.

Initially, the team installed the RECAS sonde, power, and communication systems in a dedicated area of the ice surface above
a subglacial lake. Prior to deployment, we sterilized all downhole components using a combination of chemical washing,
hydrogen peroxide vapor, and ultraviolet sterilization. The sterilization procedure was basically the same to the microbial
control method designed by the Lake Ellsworth Consortium for UK subglacial sampling probe (Siegert et al., 2012). Upon
activating the lower melting tip, the sonde started to melt down into the ice bed. Just as Philberth probe, the melted water was
not recovered from the hole and it refroze behind the sonde. The power-signal cable was released from the coil inside the
sonde. The coil motor slowed down the paying out of the cable in such way that the support point was above the centre of
gravity of the RECAS. Because of this, the sonde was inherently stable, and its attitude was plumb at all times. At this point,
research personnel could leave the site because the sonde operated as a fully autonomous system. The smart drilling system
measured, controlled, and adapted to conditions throughout the drilling operation. An automatically controlled diesel generator
provided power till the mission would be completed. Coded data from the RECAS were transmitted to a computer on the
surface and automatically send via satellite communication to the mainland for inspection. The connection with RECAS was
established with Secure Shell (SSH) Protocol which is commonly used for secure remote login over an insecure network.

When the sonde entered the subglacial lake, it sampled the water and examined water parameters, such as pressure,
temperature, pH, and conductivity (Table 1). The pressure and temperature sensors were custom-made on the base of standard
sensing elements, while conductivity sensor was completely developed by ourselves. The pH sensor was a standard sensor of
AMT-6000 type consisting of glass electrode and reference electrode. Upon completing the sampling and monitoring, the coil
motor was activated, and the top thermal tip was powered. The recovery of the sonde to the surface began with spooling of the
cable. Finally, the sonde reached the surface and was ready for service to move to the next site. Therefore, once the sonde
reached the ice base, the subglacial lake remained isolated from the surface reducing the risk of lake contamination.

One of the key steps of the project implementation was to find the optimal relationship between the sonde length, diameter
and power consumption with the diameter and length (volume) of the used cable. Balance between space for 6-mm diameter
cable and power requirements were found optimal when 900-1000 m of cable is coiled onto the inner winch of the sonde.



Finally, the RECAS working prototype (outer diameter 180 mm, length 7.275 m, weight 330 kg, maximum power consumption of heaters 8.82 kW) with 500 m long cable inside was built and prepared for laboratory and field tests (Fig. 2).

**3. Results: field tests in Antarctica**

After numerous laboratory tests with melting tips (Li et al., 2020), cables (Zhang et al., 2021), electronic control systems (Peng et al., 2021) and heating control systems (Yu et al., 2021), the RECAS working prototype was finally prepared for Antarctic field tests in 2021-2022. The test site was selected at a point ~12 km south of the Chinese Zhongshan Station on the flank of the Dålk Glacier in East Antarctica. Even the site demonstrates cold-based condition (Talalay et al., 2021), it was thought to be enough for the principal testing of RECAS functionality.

After careful preparations, the drilling of the test hole was started on January 18 (Fig. 3). At the depth of 40.5m, we observed that the temperature sensor of the upper melting tip and load sensor worked abnormally, and the inner winch stopped working. Since the borehole was not frozen yet, the sonde was lifted to the surface for maintenance. It was found that the temperature sensor leaked, resulting in the short failure of 12V power supply circuit and the damage of inner winch control circuit board. After repairing, the sonde was lowered into the borehole again, and in 103 hours the base of the ice sheet was reached at the final depth of 200.3 m. At the base, the water samples were taken into two 360 mL sterilized sample bottles and the following parameters of melted water were recorded: pressure 1.69 MPa, temperature 6.7°C, and conductivity 0.0299 mS/cm. When sampling and monitoring were complete, the sonde began to recover itself to the surface by spooling the cable and melting overlying ice. The total surface input power during drilling was 10.11 kW, including 6.15 kW for melting tips, 2.45 kW for lateral heaters, 1.43 kW cable losses, and 0.08 kW for measurement and control system. The average downward penetration rate was 1.85 m h$^{-1}$, and the average upward penetration rate was 2.94 m h$^{-1}$. The borehole inclination was in the range of 1.1-1.6°.

**4 Discussion**

During field tests, all systems of RECAS worked in expected range of power consumption. The downward penetration rate slightly exceeded our primary estimations apparently because melting tips efficiency is higher than it was estimated. The upward penetration rate in the freshly-frozen ice was predictably higher than downward penetration rate. Tests revealed few weak places in the sonde design – sensors hermiticity, performance of cable tension sensor, evenness of the level winder moving which have to be modified in the future. In addition, plans are underway to increase the drilling depth of the sonde and enhance the means of stabilization. Moreover, we are also working to extend the capability of safe and fast drilling in firn and dusty ice.

Furthermore, we plan to penetrate the active subglacial lake covered by 540-m thick ice at the Flade Iceblink ice cap in northeast Greenland. Another potential target is subglacial Lake Qilin (~42 km in length) discovered in the world's biggest canyon system under Antarctic ice in the Princess Elizabeth Land. The lake is overlain with an average ice thickness of 3600 m. Because the lake is very deep, it is proposed to access the lake in two stages. The first stage would include drilling the pilot



hole to the depth of 3000-3200 m with the clean hot-water drilling system providing an access hole with at least 250 mm in diameter. During the second stage, the rest 400-600 m thick "ice bridge" will be penetrated by RECAS sonde. When the sonde reaches the ice sheet base, the subglacial lake remains isolated from the surface allowing to take clean water samples. During RECAS deployment, the main hole is kept open with required diameter by regular hot-water reaming.

Although dust content, salinity, temperature, and radiation vary significantly between Earth and extraterrestrial bodies, subglacial lakes in Antarctica are considered to be close analogs of subsurface layers of Mars and icy satellites (such as Titan, Enceladus, and Europa). The recent thermal probes for extraterrestrial investigations do not have well-designed system to recover probes to the surface after deployment (Bar-Cohen and Zacny, 2021). Successful RECAS tests give the good example on how extraterrestrial probes can be back in safely and reliable manner.

The RECAS sonde and its field operation are relatively cheap: it is estimated to be 10–20 times less expensive than penetration with a hot-water drilling system, while installation and operation requires only four specialist staff. The successful tests proved reliability of the new 'spider' drilling approach for subglacial lakes sampling and analyzing, and, accordingly, we believe that the RECAS sonde will begin a new chapter in subglacial lake studies in Antarctica, Greenland and other regions.

*Competing interests*. The authors declare that they have no conflict of interest.

  *Acknowledgements*. This work was supported by several Chinese government and local agencies with the main funding by the National Key Research and Development Program of the Ministry of Science and Technology of China (Project No. 2016YFC1400302). We would like to thank specialists from Polar Research Center, Jilin University, Changchun; Polar Research Institute of China, Shanghai; Nanjing Institute of Astronomical Optics and Technology, National Astronomical

Observatories, CAS, Nanjing; College of Electronics and Information, Hangzhou Dianzi University; Aerospace System Engineering Shanghai (805), Shanghai; and National Ocean Technology Center, Tianjin, who were involved in the project. The project would not be accomplished without their continuous support and valuable help.

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





**Table 1: Main parameters of RECAS onboard sensors and sampler**

| Parameters | Sensor/sampler number and type | Range of measurement/sampling | Precision |
|---|---|---|---|
| Pressure | 2 × Piezoresistive sensors OST72 | 0…3500 dbar | ±0.05% F.S. |
| Temperature | 2 × Thermistors OST71 | -5…35°C | ±0.002°C |
| pH | 2 × Glass pH electrodes | 2…11 | ±0.05 |
| Conductivity | 2 × Inductive sensors OST75 | 0…70 mS/cm | ±0.005 mS/cm |
| Water sample | 2 × Electromagnetic opening/closing bottles | 0…360 mL | NA |




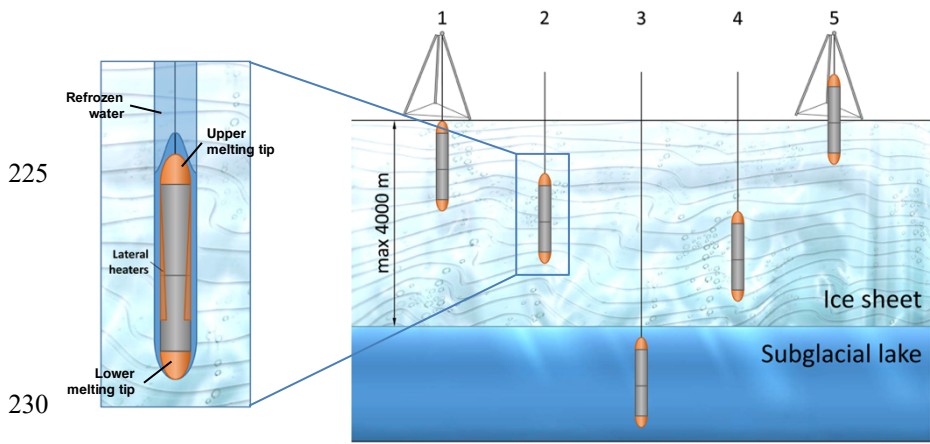



**Figure 1: RECAS subglacial access and sampling operations: 1 – activation of the sonde; 2 – drilling downwards; 3 – lake sampling; 4 – drilling upwards; 5 – arrival to the surface (modified from Talalay et al., 2014).**






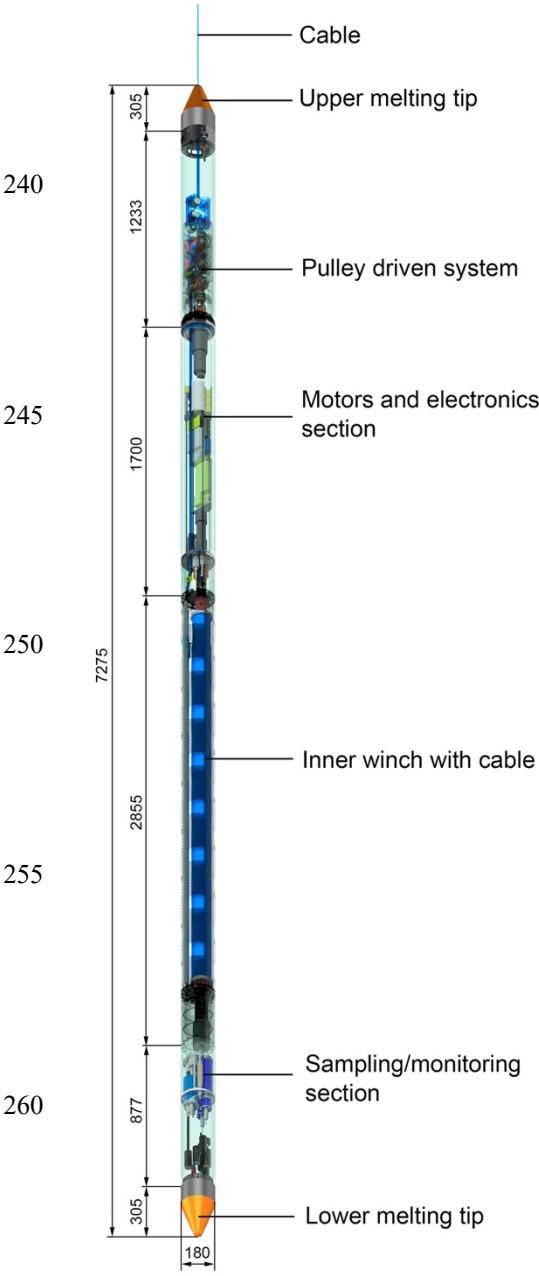

**Figure 2: 3D-model of the RECAS working prototype with 500 m long cable inside (sonde body is shown transparent; all dimensions**
265 **are in mm).**




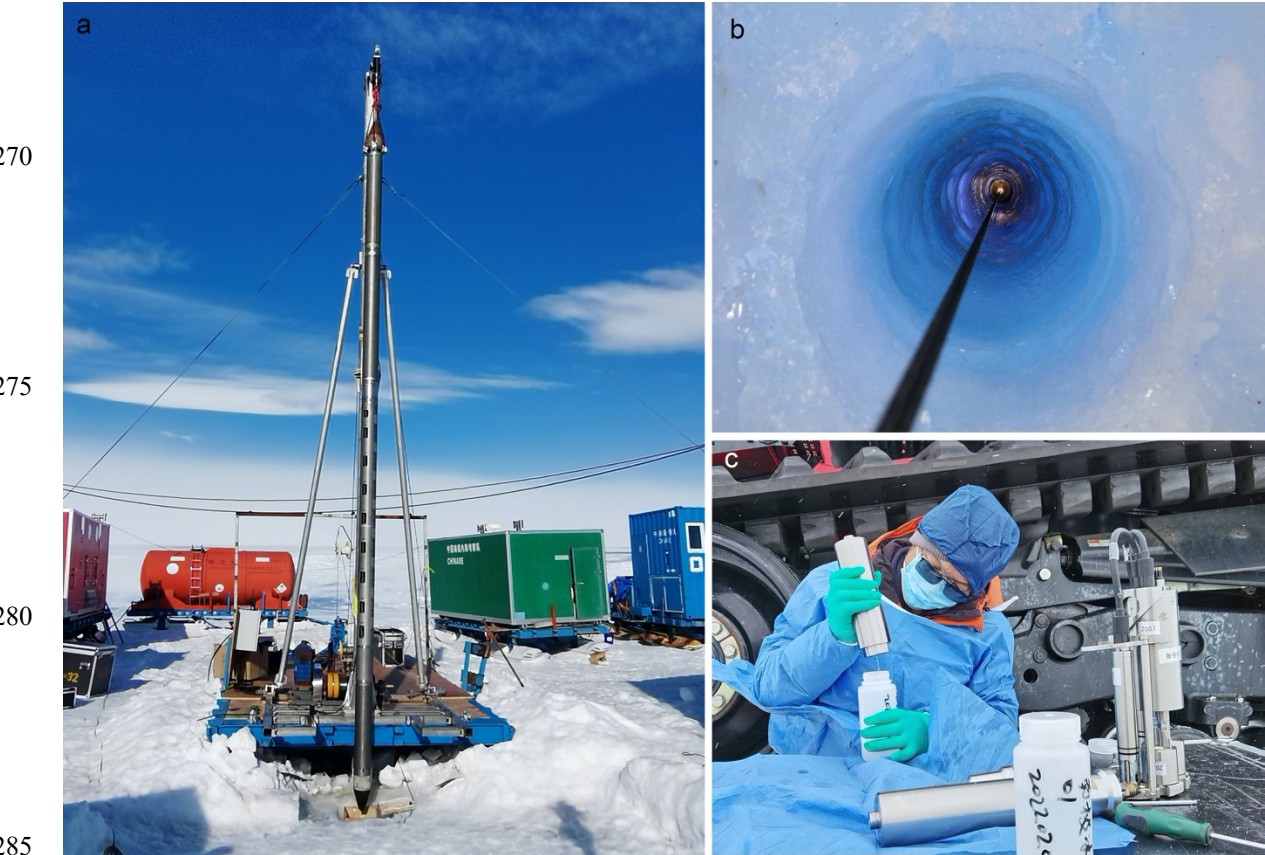

**Figure 3: RECAS field tests in Antarctica near Zhongshan Station, January 2022: (a) RECAS sonde (in the foreground) is ready for deployment; (b) Sonde on the way to the base of the ice sheet: because of the surface crevassing, meltwater drained from the borehole and down to the depth of approximately 11 m it remained dry; (c) Sample of basal ice water is poured from sterile pressurized bottles.**