# Peer review of "Brief communication: New sonde to unravel the mystery of polar subglacial lakes"

_The Cryosphere, 2022_

## Community Comment (CC1)

*The manuscript provides an interesting and well-done summary about a the novel melting probe concept (RECAS) developed by the authors and its field test results in 2022 in Antarctica.*

Thank you for your interest shown to our research.

*The devices of others are "probes" but your device is a "sonde". What is the difference that justifies another name?*

There are no any specific differences between "probe" and "sonde" in terms of freezing-in hot-point drills. We agree that the most common is the term "probe". However the term "sonde" can also be found. For example, Morton and Lightfoot (1975) alternatively used terms *meltsonde probe*, *meltsonde*, *sonde*, and *probe* to describe the concept of freezing-in hot-point drill designed by Australian Antarctic Division. From the very beginning, our drill was denoted as the sonde.

Morton BR, Lightfoot RM (1975) A prototype meltsonde probe-design and experience. Australian Antarctic Division, Department of Science, Tech Note No 14

*On page 5, in the discussion of later extraterrestrial deployments, the authors should state that the current dimensions, mass, and power consumption may be problematic and miniaturisation may be an issue ... and, well, the diesel generator will not work :)*

The words in line 138-139"*Successful RECAS tests give the good example on how extraterrestrial probes can be back in safely and reliable manner*" will be changed in the revised version of the paper as "*Successful RECAS tests give the good example on how extraterrestrial probes can be back in safely and reliable manner, although RECAS sonde cannot be applied in extraterrestrial bodies directly.*"

*The English is understandable but could be improved.*

We will improve English in the revised version of the paper.

*Page 2: "Russian team" => "a Russian team", "UK team" => "a UK team". (Dozens of "a"s and "the"s are missing in the manuscript, but they will hopefully be inserted during the generation of the proof.)*

We will carefully check and improve English grammar and style in the revised version of the paper.

*Page 3: The Philberth-type probe was proposed in "(1976)"? It reached 1005 m in 1968? So it was proposed before 1968 and maybe the original paper, in which it was proposed, should be cited instead.*

That is the common way of citation. It does not mean that the Philberth-type probe was proposed in 1976. It means that Philberth's paper was published in 1976.

*Page 3, "move in the borehole using inner cable recoiling mechanism on the same way as a spider climbs on the silk line": Well, I'm not an arachnologist, but I'm quite sure that spiders do not have such a device :) So it may be better to write "move in*

*the borehole using an inner cable recoiling mechanism similar to a spider climbing on the silk line"*

We will change the words in the revised version of the paper as suggested.

*Page 3: The changes between "the team", "we", "research personnel", "ourselves" is a bit confusing. Are they different people? Could be harmonised.*

These words will be unified in the revised version.

*Page 3, line 85: "send" => "sent"*

We will change it in the revised version of the paper as suggested.

*Page 4, "Even the site demonstrates cold-based condition ...": I don't understand this sentence.*

The initial idea was to find a site with a subglacial water system near Zhongshan Station for field testing. However, due to the impact of the COVID-19 and the overall task arrangement of the CHINARE, detailed radar topographic mapping and route exploration were not possible. Therefore, the blue ice area on the flank of the Dålk Glacier was chosen as test site, where the bedrock coring drill had been tested during the 2018/2019 season. Even there is no subglacial water under the ice sheet in the chosen site, we believe that the overlying 200 m ice is thick enough to verify the drilling/sampling functions of the RECAS sonde.

*Page 4: When the test hole was started on January 18, why was the field test 2021–2022?*

We will change the words "in 2021-2022" to "in 2021-2022 season".

*Page 5, "... can be back in safely and reliable manner": Better: "can be returned to the surface in a safe and reliable manner."*

We will change the words in the revised version of the paper as suggested.

---

## Author Response (AR1)

Dear Editor,

We are grateful to you and reviewers for fruitful comments and advices. We tried to consider all mentioned issues and hereafter explain every change made point by point. The comments are in brown, and our answers are in black. After addressing the issues raised, we believe that the manuscript would be sufficiently improved and reach the standards of The Cryosphere.

**Editor comments (Liz Bagshaw)**

Thank you for your thorough response to the reviewers comments. I now invite you to upload your revised manuscript, paying particular attention to clear communication of the results. I am happy to provide assistance if required. I would like to send the revised manuscript back out to one of the reviewers to ensure the revisions have fully addressed their concerns and that the message is clearly communicated.

We added few points into "Results: field tests in Antarctica" section according to comments of reviewers.

**Reviewer 1 (Kris Zacny)**

This is an excellent paper which illustrates development and testing of a melt probe that can be recovered.

Thank you for the kind words. Just a quick note to remind of the requirements of TC brief communication type papers (https://www.the-cryosphere.net/about/manuscript_types.html): 2–4 journal pages, 3 figures and/or tables, a maximum of 20 references, and an abstract length not exceeding 100 words. Our submitted paper already slightly exceeds the limits.

If the space allows, it might be good to provide more technical details such as structure of the cable (eg number of wires, redundant wires?, shield),

Please, see separate paper about the cable: Zhang, N., Liu, H., Talalay, P., Sun, Y., Li, N. Fan, X., Li, B., Gong, D., Hong, J., Wang, T., Liu, A., Li, Y., Liu, Y., Wang, R., Yang, Y., Wang, L.: New synthetic fiber armored cable for freezing-in thermal ice probes. Ann. Glaciol., 62(85-86), 179-190, doi.org/10.1017/aog.2020.74, (2021).

what voltage was used to sent power down to the probe (some kind of an electrical block diagram would be useful),

Please, see separate paper about the control system: Peng, S., Jiang, X., Tang, Y., Li, C., Li, X., Huang, S., Zhu, T., Shi, J., Sun. Y., Talalay P., Fan, X., Zhang, N., Li, B., Gong, G., and Yu, H.: Recoverable autonomous sonde for subglacial lake exploration: electronic control system design. Ann. Glaciol., 62(85-86), 263–279, doi.org/10.1017/aog.2021.1, (2021).

how the tensioning mechanism worked,

The separate paper "Recoverable autonomous sonde for subglacial lakes exploration: driven unit design" is currently in the works. Tension sensor itself was described in Shi J., Huang S., Wang B., Li C., Peng S., Sun Y., Talalay P., Yu H. (2021). Design and analysis of deepwater tension sensors for ice drill application. Ann. Glaciol. 62(84), 46–52. https://doi.org/10.1017/aog.2020.71

which parts of the probe were flooded, which were dry,

We will add in the revised version of the paper: "Motors and electronics were integrated in the pressure chamber while all other parts of the sonde were flooded".

was the water sample analyzed in a lab to reveal something interesting?

Because of covid situation in Shanghai, recovered samples are still in the port. Thus, analyzing of the sampled water did not start yet.

**Reviewer 2 (Bernd Dachwald)**

The manuscript provides an interesting and well-done summary about a the novel melting probe concept (RECAS) developed by the authors and its field test results in 2022 in Antarctica.

Thank you for your interest shown to our research.

The devices of others are "probes" but your device is a "sonde". What is the difference that justifies another name?

There are no any specific differences between "probe" and "sonde" in terms of freezing-in hot-point drills. We agree that the most common is the term "probe". However the term "sonde" can also be found. For example, Morton and Lightfoot (1975) alternatively used terms meltsonde probe, meltsonde, sonde, and probe to describe the concept of freezing-in hot-point drill designed by Australian Antarctic Division. From the very beginning, our drill was denoted as the sonde.

Morton BR, Lightfoot RM (1975) A prototype meltsonde probe-design and experience. Australian Antarctic Division, Department of Science, Tech Note No 14

On page 5, in the discussion of later extraterrestrial deployments, the authors should state that the current dimensions, mass, and power consumption may be problematic and miniaturisation may be an issue … and, well, the diesel generator will not work :)

The words in line 138-139 "Successful RECAS tests give the good example on how extraterrestrial probes can be back in safely and reliable manner" will be changed in the revised version of the paper as "Successful RECAS tests give the good example on how

extraterrestrial probes can be back in safely and reliable manner, although RECAS sonde cannot be applied in extraterrestrial bodies directly."

The English is understandable but could be improved.

We will improve English in the revised version of the paper.

Page 2: "Russian team" => "a Russian team", "UK team" => "a UK team". (Dozens of "a"s and "the"s are missing in the manuscript, but they will hopefully be inserted during the generation of the proof.)

We will carefully check and improve English grammar and style in the revised version of the paper.

Page 3: The Philberth-type probe was proposed in "(1976)"? It reached 1005 m in 1968? So it was proposed before 1968 and maybe the original paper, in which it was proposed, should be cited instead.

That is the common way of citation. It does not mean that the Philberth-type probe was proposed in 1976. It means that Philberth's paper was published in 1976.

Page 3, "move in the borehole using inner cable recoiling mechanism on the same way as a spider climbs on the silk line": Well, I'm not an arachnologist, but I'm quite sure that spiders do not have such a device :) So it may be better to write "move in the borehole using an inner cable recoiling mechanism similar to a spider climbing on the silk line"

We will change the words in the revised version of the paper as suggested.

Page 3: The changes between "the team", "we", "research personnel", "ourselves" is a bit confusing. Are they different people? Could be harmonised.

These words will be unified in the revised version.

Page 3, line 85: "send" => "sent"

We will change it in the revised version of the paper as suggested.

Page 4, "Even the site demonstrates cold-based condition ...": I don't understand this sentence.

The initial idea was to find a site with a subglacial water system near Zhongshan Station for field testing. However, due to the impact of the COVID-19 and the overall task arrangement of the CHINARE, detailed radar topographic mapping and route exploration were not possible. Therefore, the blue ice area on the flank of the Dålk Glacier was chosen as test site, where the bedrock coring drill had been tested during the 2018/2019 season. Even there is

no subglacial water under the ice sheet in the chosen site, we believe that the overlying 200 m ice is thick enough to verify the drilling/sampling functions of the RECAS sonde.

Page 4: When the test hole was started on January 18, why was the field test 2021–2022?

We will change the words "in 2021-2022" to "in 2021-2022 season".

Page 5, "… can be back in safely and reliable manner": Better: "can be returned to the surface in a safe and reliable manner."

We will change the words in the revised version of the paper as suggested.

**Reviewer 3 (Paul Anker)**

Summary

This paper is a good general summary of the progress made of a novel sonde instrument, its current capabilities and recent successes in field testing. It is great to see progress being made in this direction with due consideration of the unique aspects of subglacial lake access.

Thanks for your kind words about our progress. We believe that the sonde will be a good alternative instrument to study polar subglacial lakes in the future.

The methods section is a little confusing as it is largely written in the past tense following an idealised deployment.

Originally the methods section was written in the present tense. However, the English editor converted all description into the past tense. We will consider language polishing before submission of the revised version.

Maybe a more detailed and more heavily annotated diagram of the deployment and recovery process could replace much of the more prosaic descriptions of the deployment? Personally I would like to see a diagram of the instrument itself, similar to that found in the associated paper on electrical

We will add additional notes on Fig. 2.

Line 69: Paragraph above this point should be in the introduction

We will move this paragraph to introduction in the revised version of the paper.

Line 72: What mechanism is actually used, is it anything like a spider climbing?

At the very beginning we compared four different driving mechanisms. Finally, we chose the structure of the inner winch, in which three different motors are used to drive capstan, drum

with the cable and level winder as shown in the figure below. The mechanism is really quite similar to the spider climbing.

[Figure]

Line 76: How closely to deployment was sterilisation carried out? Was there any sampling of sonde surfaces before and after deployment to verify cleanliness? Any background measurements of the camp/snow to help identify potential sources of contamination?

The sterilization procedure was basically the same to the microbial control method designed by the Lake Ellsworth Consortium for UK subglacial sampling probe (Siegert et al., 2012). Before equipment was transported to Antarctica, the RECAS sonde was disassembled into individual easy-to-clean parts or modules and sterilized using a combination of chemical wash, autoclaving, HPV, and UV in China. Then RECAS parts were assembled into independent modules in the clean room and packed into vacuum bags for shipment. The deployment procedure did not include assessment to provide data for control or verification of the cleanliness in the field. However, snow samples were collected at the site for microbiological and chemical analysis.

Line 82: Was the system left entirely unsupervised? What was the reasoning behind this? Was there remote real time monitoring available? Is there redundancy in the system to manage problems if they occur i.e. power downs?

Sampling of a relatively shallow subglacial lakes (500-1000 m) can be conducted during the summer season (1-2 months) and the system can be supervised by personnel. However, exploration of a deeply buried subglacial lake requires 4–6 months. In this case, it was proposed that the research personnel leave the site after deployment and the sonde operates as a fully autonomous system. The uninterrupted power supply system was designed to provide power supply for at least 6 months in unmanned mode. The status parameters of RECAS sonde and auxiliary systems are regularly sent to monitoring center in China through the iridium system. The remote personnel can not only monitor the drilling process in the field but also send commands to intervene in the process at any time.

Line 85: Where is 'mainland'? Jilin or still in Antarctica?

"Mainland" will be changed to "China".

Line 86: What sort of control does this allow? Real time access or a subset of data transmitted periodically?

The communication architecture of the remote control system is shown in the figure below, which is composed of the Antarctic industrial computer, the Iridium satellite, the Iridium ground gateway and the domestic server. The Antarctic industrial computer and domestic server are configured with keys and SSH access environment to establish a secure data transmission channel. The Antarctic industrial computer directly accesses the IP (static) and port of the domestic server, and uses the Secure Copy (SCP) command in the SSH protocol to copy the field data to the server. At the same time, a reverse connection channel was established using SSH -R command to map the communication port of the Antarctic industrial computer to the server, so that the server can issue control commands to the Antarctic industrial computer by accessing the local mapped port, so as to solve the problem that the IP address provided by the iridium service in Antarctic is not fixed. Based on a communication link established according to this structure, the data transmission between the Antarctic industrial computer and the domestic server is safe, simple and reliable. Except for the transmission rate limited by the Iridium bandwidth, there is no difference between the data interaction with the two computers in the local area network. The main parameters of RECAS and the uninterrupted power supply system will be packed and sent to monitoring center in China at regular intervals (>1 min).

[Figure]

Line 87: Be more explicit that a subglacial lake wasn't actually entered during this test. Or am I wrong and you did actually enter a lake at 200m?!

Please, pay attention to the lines 100-101: "Even the site demonstrates cold-based condition (Talalay et al., 2021), it was thought to be enough for the principal testing of RECAS functionality." There is no subglacial water at the site, we collected meltwater from the basal ice at the boundary of the ice sheet with bedrock.

Line 88: Was this data internally logged and/or returned to the surface? What proportion of data collected was returned to the surface?

We have two storage modes for CTD data, one is directly stored internally (update data every 2 seconds), and the other is transmitted to the surface in real time (update data every 5 seconds). The two modes can be switched under surface or remote control. For the working mode of RECAS sonde, we do not think that it is necessary to set too high data collection frequency.

Line 91: Some detail of how the spooling works might be nice here, if not covered earlier in the paper.

We will add into revised version of the paper: "The sonde is raising and lowering using dual wheel capstan driving mechanism. Three different motors are used to drive capstan, drum with cable and level winder. The rotation of capstan motor and corresponding moving speed of the sonde is automatically controlled depending on the signal from the load sensor. The rotation of the drum's motor is also automatically controlled according to the signal from the tension sensor installed between capstan and drum in order to keep the cable tension in the predetermined range."

The separate paper "Recoverable autonomous sonde for subglacial lakes exploration: driven unit design" is currently in the works. TC brief communication type papers have limits of 2–4 journal pages, 3 figures and/or tables, and a maximum of 20 references. Our submitted paper already exceeds the limits, so we have no way to give detailed description of the spooling mechanism here.

Line 106: Some examples of the preparations that are specific to this instrument would be interesting here

Preparations included the routine activities like camp construction, sonde assembly and debugging, etc.

Line 110: How was the base of the ice stream recognised by the sonde?

The base of the ice was recognized at the point when RECAS penetration ultimately stopped. The final depth was about two meters deeper than the depth of the IBED borehole drilled to the bedrock by electromechanical drill ~60 m to the north-east in 2019 (Talalay et al., 2021).

Line 111: Was the sampling system triggered automatically or manually?

The sampling system was triggered manually.

Line 113: How does the lay up of the cable on the spool work? How is it monitored? Is the top of the sonde identical to the bottom? Does the cable penetration through the heated tip have any effect on melting on recovery to the surface?

Coiling/recoiling mechanism is shortly described above. We cannot monitor cable laying – everything is going on automatically. Top and bottom thermal tips are totally identical except for the central hole for cable in the top thermal tip. During lab tests under equal conditions, the bottom thermal tip drilled faster in 1.2 times than the top thermal tip (see Fig. 9a in Li et al., 2020). This is because the top thermal tip has an unheated central hole that slows down the rate of penetration. However, during field tests in Antarctica the average downward penetration rate was 1.6 times lower than the upward penetration rate. This is because drilling in the freshly frozen 'warm' ice is easier than in the cold glacier ice.

Line 116: Any knowledge of the temperature profile through the ice column? Can a temperature profile be recovered from the probes' measurements?

Ice temperature was measured in the IBED borehole (see figure below: blue markers – measured data, red line – approximated temperature profile). Temperature at the surface is -11.4 °C, and temperature at the ice sheet base is -4.8 °C (data are not published yet).

The sonde is equipped by four temperature sensors: two at the melting tips, one in the sampling/monitoring section and another one in the motors/electronics section. All of them are not intended to measure in situ ice temperature during drilling.

[Figure]

Line 134: I know this is for future work, but I would be interested to know if there are plans to protect the hose and instrument cable from tangling together whilst hot water reaming 2400m+ borehole.

We agree that tangling of the RECAS cable and the reamer hose would be a concern. As the first step, we plan to test anti-wrap fixture suggested by Blake and Price (2002, Fig. 3).

**References**

Blake, E.W., Price, B. (2002). A proposed sterile sampling system for Antarctic subglacial lakes. Mem. Natl. Inst. Polar Res., 56, 253-263.

Li, Y., Talalay, P.G., Sysoev, M.A., Zagorodnov, V.S., Li, X., and Fan, X. (2020). Thermal heads for melt drilling to subglacial lakes: Design and testing. Astrobiology, 20(1), 1-15.

Siegert, M.J. et al. (2012). Clean access, measurement, and sampling of Ellsworth Subglacial Lake: A method for exploring deep Antarctic subglacial lake environments, Rev. Geophys., 50, RG1003.

Talalay, P., Li, X., Zhang, N., Fan, X., Sun, Y., Cao, P., Wang, R., Yang, Y., Liu, Y., Liu, Y., Wu, W., Yang, C., Hong, J., Gong, D., Zhang, H., Li, X., Chen, Y., Liu, A., and Li, Y. (2021). Antarctic

subglacial drill rig. Part II: Ice and Bedrock Electromechanical Drill (IBED). Ann. Glaciol., 62(84-85), 12-22.

On behalf of the authors,

Pavel Talalay

Xiaopeng Fan

---

## Author Response (AR2)

Dear Liz and Bernd,

Thank you very much for polishing the paper, comments and suggestions. We hope that the current version of the paper would avoid obscurities and grammatical irregularities and reaches TC requirements.

Regarding to Bernd's comments:
1. Page 2, line 36: From the wording, it is not clear that the lunar rocks gave insight into the origin and evolution of our planet, not subglacial samples (at least not into the origin). A better wording may be: "... comparable to the lunar rocks, which gave insight ...".
Corrected.

2. Page 2, line 56, "The polar engineering community continues to work on the development of the new concepts and technologies for sampling subglacial lakes". This sentence refers to the time after the latest field tests. But the references are from 2002 and 2007???
Reformulated as following: "In addition, there have been several proposals for sampling subglacial lakes that did not go further than preliminary conceptual level (Blake and Price, 2002; Fleckenstein and Eustes, 2007)."

3. Page 3, line 79: "do" => "does"
Corrected.

4. Page 3, line 90: "lakes" or "a lake", but not "a lakes"
Corrected.

5. While the English is understandable, it still needs to be improved. For example, quite some "the"s and "a"s are missing.
The paper was checked by professional English editors two times (before first and second submissions) but perhaps not good enough. Scientific editor, Liz Bagshaw, helped us to check and to polish the current version of the paper.